# Enabling Composite Optimization through Soft Computing of Manufacturing Restrictions and Costs via a Narrow Artificial Intelligence

**Markus Edwin Schatz** [†]

Airbus Defence and Space GmbH, Claude-Dornier-Strasse, 88090 Immenstaad, Germany;
schatz@llb.mw.tum.de; Tel.: +49-7553-8855
† Current address: Weildorfer Hardt 14, 88682 Salem, Germany.

**Abstract:** In industry, manufacturing has a huge imprint on structural design; which particularly holds for composites. This is caused by complex interaction of geometry, process parameters and material quantities e.g., fiber orientation. This interaction yields a wide variety of feasible designs, which severely differ in costs and structural performance, measured in mass, stiffness and strength. In order to cope most effectively with this complexity, this paper discusses a weak artificial intelligence, emulating human expertise on composite manufacturing. This approach is extended such that the used knowledge-based system is capable of providing a reason for having determined a certain level of manufacturing effort. Moreover, this extension also provides advice pointing into the direction of optimal improvement. These novelties may be used during designing, optimization and post-processing. These three cases are herein discussed by applying it onto an automotive structure.

**Keywords:** composite analysis; structural design optimization; soft computing; braiding process; preform optimization; modeling manufacturing restrictions; quantifying production effort; knowledge-based system; narrow artificial intelligence

## 1. Introduction

### 1.1. Motivation

Enhancing composite design and structural composite optimization through the consideration of manufacturing costs and restrictions was the primary source of motivation. This is a key to success since composite manufacturing not only introduces restrictions on shape, material and geometry quantities, but moreover decisively determine costs and structural performance. The latter shall be understood as ratio of stiffness and/or strength to mass. Clearly, considering manufacturing restrictions and costs at an early stage of design development results in a huge potential in designing lightweight structures that are also producible at low or moderate expenses.

The author laid some groundwork in this matter with [1]. The research work presented herein, is intended to extend this foundation, such that it can be used more effectively in the context of composite optimization. Because of its huge potential in being cost efficient, whilst still enabling performant composite lightweight structures and its inherent process complexity, braiding has been considered as baseline process to demonstrate the applicability of the approach. An illustrative sketch is given with Figure 1. More information is to be found in [2].

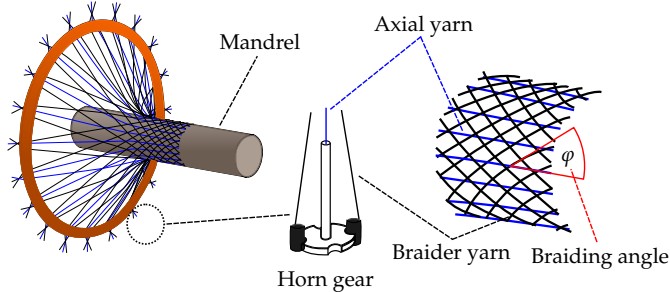

**Figure 1.** Schematic sketch of preforming technology braiding.

## 1.2. Brief Literature Review on Capturing Manufacturing Restriction and Costs

After reading basic literature [2,3], one can clearly conclude that manufacturing restrictions have a considerable influence on the design of composites. As outlined, this paper aims at methods capturing manufacturing restrictions and costs, such that they can be considered within an optimization frame. With Figure 2, the three most frequently used approaches by far are illustrated: direct process simulation, analytical bounds and soft computing.

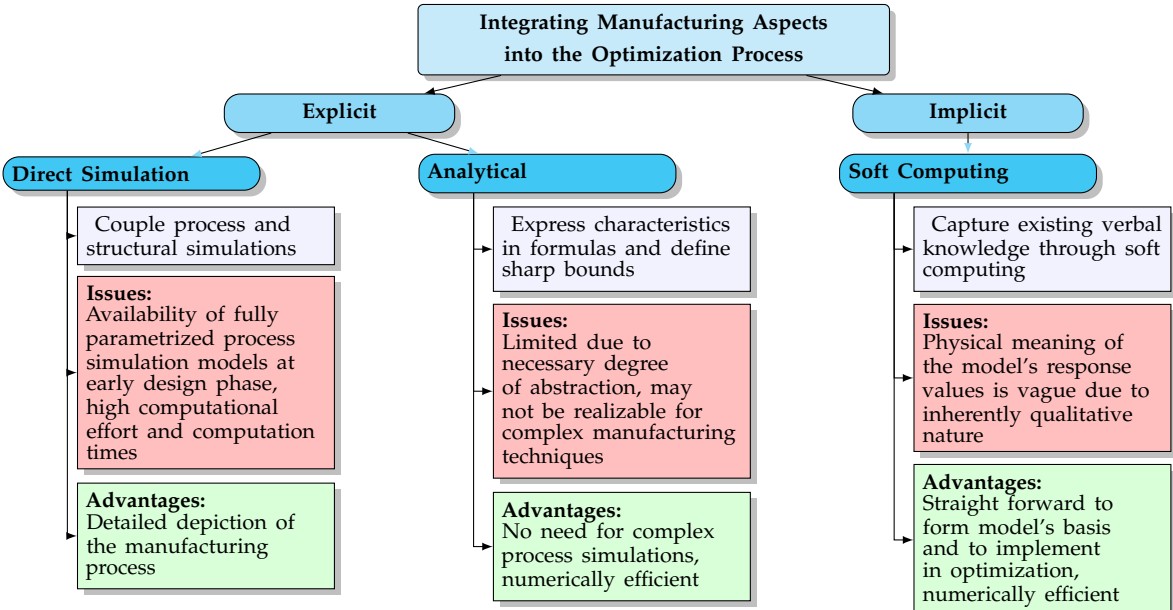

**Figure 2.** The three most frequent cases of capturing manufacturing aspects if they are to be introduced into the optimization process.

Pickett et al. [4] completed a remarkable piece of research on simulating the braiding process using a finite element method. This unfortunately came at certain computation costs, since braiding evidently does involve a strong interaction of fiber yarns, which is challenging to consider. For other composite manufacturing techniques such as injection molding, more research on design optimizations including process simulations exist. One example is given by Chen et al. [5], who captured the imprint of the four most relevant process parameters on shear layer thickness. With the work of Mackerle [6], an overview on composite manufacturing simulations between 1985 and 2003 is given. After reading this and considering the research on this matter, direct simulations tend to require a high amount of expenses. On top of this, it might be time-consuming and difficult to set up. This especially holds for composite manufacturing techniques such as braiding.

Therefore, the more promising approach by far is to capture manufacturing aspects via analytics. In literature, there are many works related to this. Wang et al. [7] were among the first to consider manufacturing constraints within an optimization frame. Henderson et al. [8] and Ghiasi et al. [9]

shared a similar approach and did also experienced that objectives arising from production and mechanics tend to be of a conflicting nature. Hence, minimum mass and minimum cost designs are obviously two different ones. Even though their research work did lift composite optimization to a higher significance, it is most suitable for manufacturing techniques allowing analytic abstraction and therefore might be of a rather simple nature.

For that reason, the third branch of modeling approaches did emerge, thus capturing manufacturing aspects via soft computing. This terminology already implies that, instead of founding the modeling approach on hard analytics, soft numbers and relationships are captured. This is closer to human perception by far, which often elaborates based on verbal and qualitative information. It is thereby able to intuitively cope with uncertainty and fuzziness. This was impressively shown by Iqbal [10], where a knowledge-based system was used to capture relevant aspects of milling process in an efficient manner. Zhou et al. [11] and Huang et al. [12] did pick up this approach so as to build narrow intelligent machines for capturing essential process aspects. Those three contributions impressively show how soft computing leverages optimization to a whole new level of holisticity and therefore significance, even in light of complex manufacturing processes being hard to model via classical approaches. Wehrle [13], for instance, used this approach to cover uncertainties associated with material parameters. The basis for these research works actually trace back to Shortliffe [14] and the birth of a knowledge-base system, which were back then referred to as expert systems. These systems can be interpreted as illustrated with Figure 3.

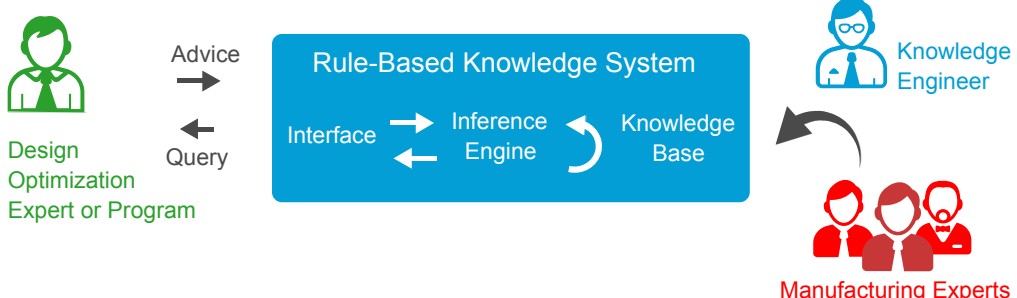

**Figure 3.** Knowledge-based system and its interface to optimization and experts via a knowledge engineer.

Thus, the core elements of these knowledge-based systems most frequently are interface to user or program (i.e., optimization), interference engine and the actual knowledge-base [15]. As can be observed, these systems are hence not intended to dynamically learn, but instead a knowledge engineer transfers expertise into a rule based knowledge-base. More details on this are to be found in [1]. For further reading on current methods and emerging trends in soft computing, consult Hajela [16].

*1.3. Application Cases*

As outlined in Section 1.1, the motivation for this paper was the efficient analysis and optimization of composite structures in general and in particular of braided ones. With the following Figure 4, a practical example of interest is given. This example was already discussed at length with [17], where it became obvious how manufacturing effort modeling enables structural design optimization. It was actually so efficient that the complete Pareto frontier was resolved using a gradient-based approach. However, the approach was limited in a sense that it provided only single values per query and reasoning or advising during post-processing was not available. This is the major goal of this paper. In addition, another structure is considered; an A-pillar reflecting automotive applicability.

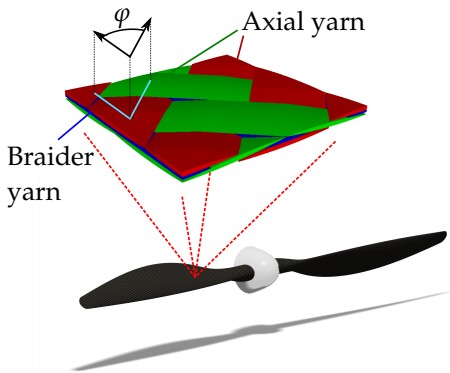

**Figure 4.** Propeller structure and used meso-model [17].

### 1.4. Used Material Properties

As the braiding process is unchanged, the used material model of [17] is used again and depicted with Figure 5. As one can observe, the material varies in-between 30 and 80 MPa in extrusion or braiding axis and matches quite well with experimental values; see the green dots.

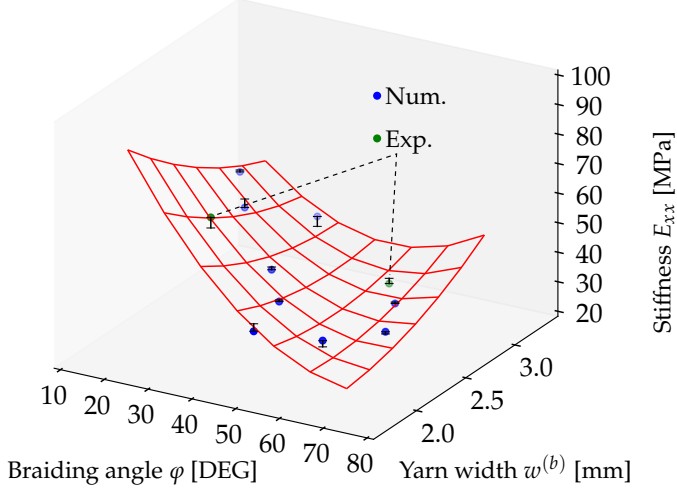

**Figure 5.** Stiffness $E_{xx}$ over braiding angle and yarn width [17].

Up to now, neither coefficient of thermal expansion nor thermal conductivity were considered. However, the approach used would allow homogenization for those two as well. Moreover, this could be realized with ease, since the constitutive law for those thermal quantities is of tensor order two instead of four and does not contain off-diagonal entries.

## 2. Modeling Manufacturing Effort

### 2.1. Quantification via Measure Effort

As outlined above, the idea is to transfer common knowledge of experts on a manufacturing technique of interest into a knowledge-base so as to provide a depictive example, Figure 6 illustrates how the two dimensions' complexity of lengthwise geometry and complexity of profile influence producibility in a qualitative manner.

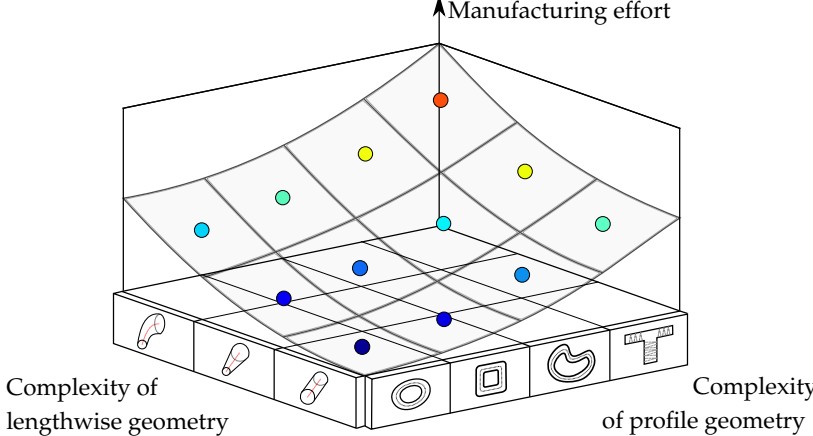

**Figure 6.** Manufacturing effort over complexity of profile and lengthwise geometry [1].

In order to have a general measure, effort ranging from zero to one hundred percent has been defined. This is advantageous, since it is independent of any business model including country or company depended variables. Moreover, it helps to generalize producibility, which is for instance defined differently for the automotive industry because of tight cost constraints, compared to space industry. Last but not least, this approach proved to be superior, due to the smooth response surfaces it yields. This continuity is relevant when choosing optimization algorithms. Figure A2 in Appendix B displays this continuity, allowing the use of gradient-based optimization algorithms. This is key for numerical performance if a vector optimization problem is formed, as discussed in detail in [17].

### 2.2. Extension via Reasoning and Advising

The following Figure 7 sketches the principal of the developed approach. Thus, multiple rules are defined so as to correlate braiding angle $\varphi$ in tandem with braid yarn width $w$ to manufacturing effort $e$. In braiding, the yarn width is determined via the circumference $U$, number of bobbins $n_B$ and the braiding angle $\varphi$, as given by Equation (1):

$$w = \frac{2U}{n_B} \cos \varphi. \tag{1}$$

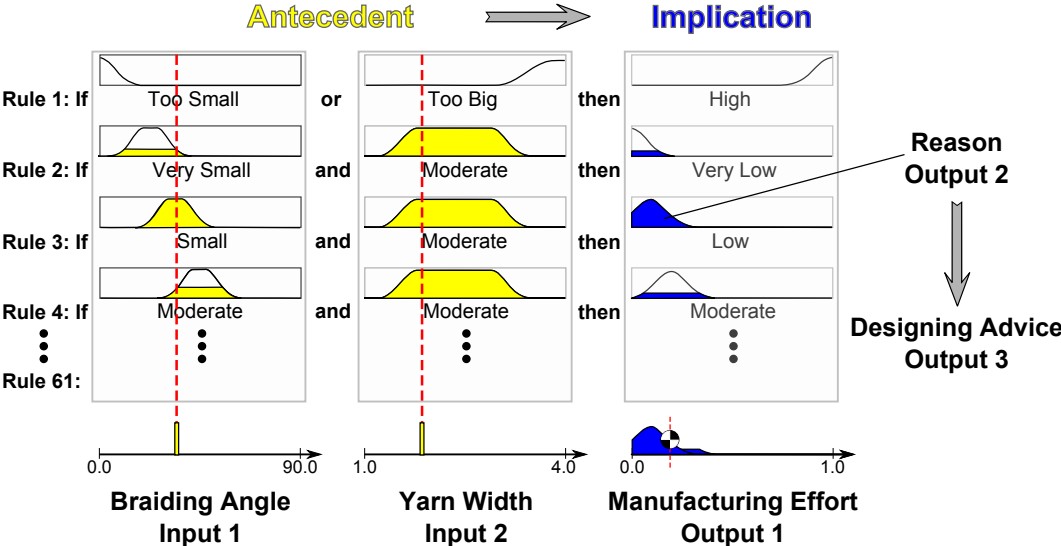

**Figure 7.** Projection of braiding angle $\varphi$ and yarn width $w$ to manufacturing effort $e$.

Rule number one for instance reflects the case, that yarn width $w$—determined by number of bobbins and braiding angle $\varphi$—is too great. Thus, the preform would open at the same points as given by Figure 8b. This rule is 100% associated with manufacturing effort, since this configuration would actually mean scrap. Moreover, because of the OR-link in-between the two fuzzy variables braiding angle and yarn width, it also accounts for the case that the braiding angle is zero.

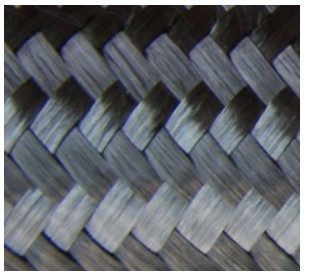
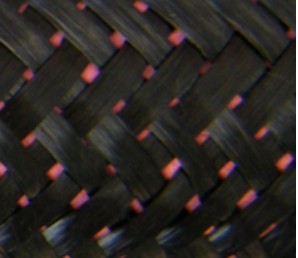

(**a**) perfect braid                          (**b**) opened braid

**Figure 8.** Two different braid configurations for a braiding angle $\varphi$ of 45°.

Rules 2–7 reflect all feasible solutions, which only differ in take-up speed and therefore translate into costs via production time. This is because a braiding angle tending to 20 degrees yields a minimum production time, whereas 90 degrees correlates to an infinitely long production time. As can thus be observed, the rules basically allow the combination of fuzzy variables having values like *Too Small* or *Moderate*. Thus, one only needs to translate a given value into those fuzzy sets. This is depicted at the bottom of Figure 7, where the red line depicts the sharp input values, e.g., 35° and 2.2 mm. The fill of the fuzzy sets in yellow reflect to what extent the sharp value matches. Hence, 35° does belong to the fuzzy set 100% *Small*. This first step is called the antecent. Thereafter, each rule is evaluated on how active it is. The most common approach is to search the minimum weight of all input fuzzy sets for an AND-link and the maximum weight for an OR-link, respectively. This is called implication. Ultimately, all fuzzy sets are aggregated to an output fuzzy set. A sharp number can then be derived by computing its center of area. Clearly, in Figure 7, rule number three is the most dominating one by far.

With Figure 9, a generalized so-called Mamdani fuzzy inference system is given. Fuzzy sets are illustrated via blue lines, where each would represent a verbal variable. Implication is given via red fill and aggregation with green.

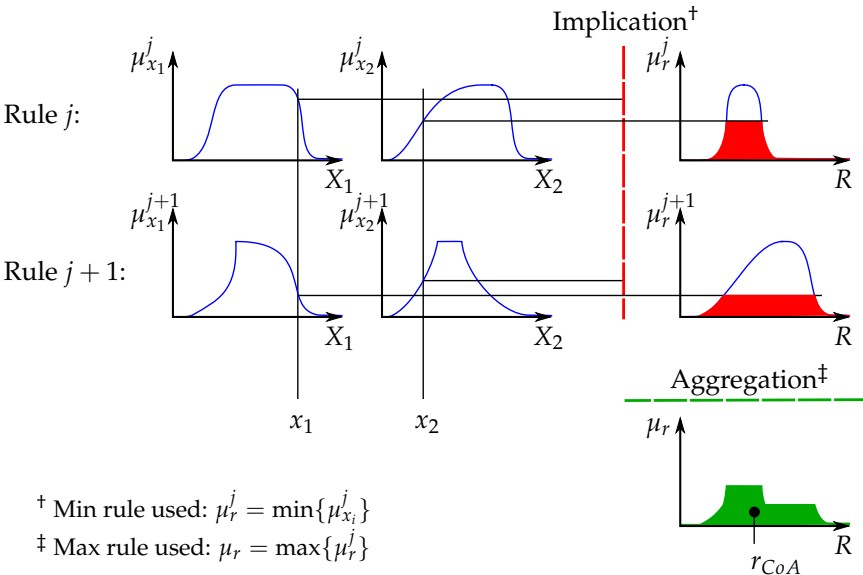

**Figure 9.** A general Mamdani fuzzy inference system.

If therefore $\mu_{x_i}^j$ represents the fuzzy set's weight of input variable $i$ and rule $j$, the following equation set (2) defines the output weight of rule $j$ for an AND and OR link:

$$\mu_r^j = \begin{cases} \min\{\mu_{x_i}^j\} & \text{AND implication,} \\ \max\{\mu_{x_i}^j\} & \text{OR implication.} \end{cases} \tag{2}$$

This weight (2) marks the result of implification and thus describes the level of participation per rule. With this, one can determine the final output fuzzy set (green surface in Figure 9) via aggregation. This is either realized by a maximum or sum approach, as given by (3). The final step in a Mamdani FIS is given by defuzzification, which is most frequently realized by computing the center of area via the first static moment (4):

$$\mu_r = \begin{cases} \max\{\mu^j\} & \text{MAX aggregation,} \\ \sum_j \mu^j & \text{SUM aggregation,} \end{cases} \tag{3}$$

$$r_{CoA} = \frac{\int_R r \cdot \mu_r^j(r)\,dr}{\int_R \mu_r^j(r)\,dr}. \tag{4}$$

Such a general Mamdani FIS can be extended by querying for arguments that determined the final output $r_{CoA}$ the most. Knowing this allows the knowledge engineer to a priori implement reasons and advice for each rule and each fuzzy input set of a given rule. Thus, one first has to query for the dominating rule as given by Equation (5):

$$j_{\text{active}} = \begin{cases} \arg\max\{\mu^j\} & \text{MAX aggregation,} \\ \max\{\mu^j\} & \text{SUM aggregation.} \end{cases} \tag{5}$$

Knowing the most dominating rule $j_{\text{active}}$—here referred to as active—one can fetch the most dominating fuzzy input set as illustrated by (6). Hence, the active fuzzy input set $i_{\text{active}}^j$ is defined by querying for arguments of the maxima for OR rule and of minima for AND, respectively.

$$i_{\text{active}}^j = \begin{cases} \arg\min\{\mu_{x_i}^{j_{\text{active}}}\} & \text{AND implication,} \\ \arg\max\{\mu_{x_i}^{j_{\text{active}}}\} & \text{OR implication.} \end{cases} \tag{6}$$

With Table 1, an overview of the implemented reasons and advice of rule number one and three of the example as discussed above and illustrated with Figure 7 is provided. As outlined before, rule number one reflects the case of a scrap braid preform. It, for instance, reflects the case of braid opening as illustrated via Figure 8b and is for a given shape and defined braiding angle defined by yarn width $w$. This case is represented by the second row of Table 1, where the knowledge engineer has defined the reason as well as advice for the known scenario.

**Table 1.** Implemented reasons and advice for rules number one and three.

| $j_{\text{active}}$ | $i_{\text{active}}^j$ | Reason | Advice |
|:---:|:---:|:---:|:---:|
| 1 | 1 | In rule *BraidOpening*: *Angle* is *Too Small* | Reduce take-up speed |
| 1 | 2 | In rule *BraidOpening*: *Yarn Width* is *Too Small* | Increase take-up speed, more filaments, greater yarn size et cetera |
| 3 | 1 | In rule *AllGood*: *Angle* is *Moderate* | Increase take-up speed (process time) |
| 3 | 2 | In rule *AllGood*: *Yarn Width* is *Moderate* | Increase take-up speed (process time) |

In Appendix C, the reader may find an overview on the implemented knowledge-base in form of a network graph (see Figure A3). The above Figure 7 represents one of these elements, namely the $\varphi$-$w$-FIS of Figure A3.

## 3. The Demonstration Structure: A-Pillar

Before discussing all findings, the demonstration example in the form of the A-pillar structure is introduced first. The A-pillar is part of the Roding convertible as illustrated by Figure 10.

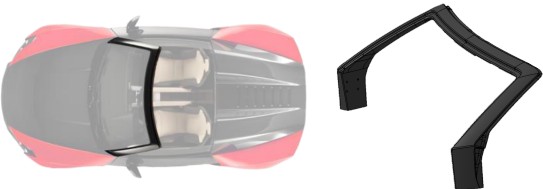

**Figure 10.** The demonstration example A-pillar.

This structure is clearly determining the overall torsional and bending stiffness of the sports car and thereby pinning down its driving dynamics. This is why it needs to fulfill certain requirements on stiffness as given by the three unit load cases, as illustrated by Figure 11b. On top, the A-pillar also needs to fulfill requirements arising from crushing it, once the car overturns (see Figure 11a).

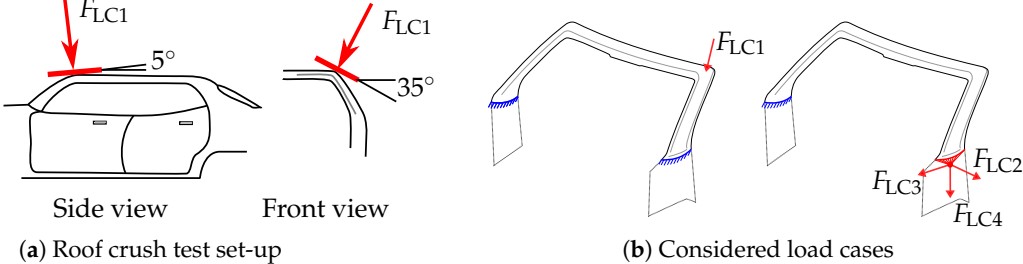

(**a**) Roof crush test set-up　　　　　　　　　　　(**b**) Considered load cases

**Figure 11.** Load cases for the A-pillar design task.

## 4. Application of Approach and Usage of Its Novelties

### 4.1. Embedding into Optimization Frame

The manufacturing effort model (MEM) can moreover be used to augment composite optimization. This is visualized by Figure 12, where structural mechanics responses computed by FEM are coupled with the ones defined by the MEM.

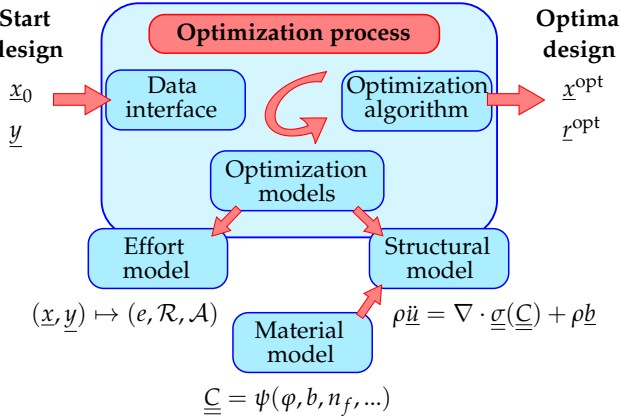

**Figure 12.** General optimization process as by [17].

This coupling of mechanical responses like stiffness, mass and failure indices with manufacturing effort can be represented by the following optimization problem (7). This optimization problem states that the objective $f$ is to be minimized by a change of design variables $\underline{x}$, while still fulfilling all constraints given by $\underline{g}$:

$$
\begin{aligned}
\underset{x \in \chi}{\text{minimize}} \quad & f, \\
\text{subject to} \quad & g_l \leq 0, & l = 1, \ldots, n_{\text{IC}}, \\
\text{with} \quad & \chi = \{x_j \in \mathbb{R}^{n_{\text{DV}}} : x_j^l \leq x_j \leq x_j^u\}. & j = 1, \ldots, n_{\text{DV}}.
\end{aligned}
\tag{7}
$$

In Appendix A, the parametrization and design space of the A-pillar optimization problem is given. The inequality constraints $\underline{g}$ were here defined by:

$$g_1(\underline{x}) = 1 - \frac{K_{xx}}{K_{xx,\min}} \quad \text{Stiffness requirement in } x \text{ direction,}$$

$$g_2(\underline{x}) = 1 - \frac{K_{yy}}{K_{yy,\min}} \quad \text{Stiffness requirement in } y \text{ direction,}$$

$$g_3(\underline{x}) = 1 - \frac{K_{zz}}{K_{zz,\min}} \quad \text{Stiffness requirement in } z \text{ direction,}$$

$$g_4(\underline{x}) = \frac{u_{\text{LC1}}}{u_{\text{LC1,max}}} - 1 \text{ Limit on intrusion for roof crush case,}$$

$$g_{5+s}(\underline{x}) = \frac{\mathcal{FI}_{\text{LC1},s}}{0.9} - 1 \text{ No failure for the roof crush case } \forall s \in \{1, \ldots, n_{\text{Sec}}\}.$$

Mass and manufacturing effort are conflicting objectives, since reducing mass clearly comes at production expanses and vice versa. Therefore, the objective definition needs to reflect this. Equation (8), where $\alpha_m$ and $\alpha_e$ allow weighting either mass or effort does thereby outbalance conflicting objectives, if both responses are normalized by their individual optimum:

$$f_d = d(m, e) \qquad \text{(compromise objective)}, \tag{8}$$

$$
\begin{aligned}
\text{with} \quad & d(m, e) = \|(\hat{m}, \hat{e})\|_2, \\
& \hat{m} = \frac{\alpha_m (m - m^{\text{opt}})}{m^{\text{opt}}}, \\
& \hat{e} = \frac{\alpha_e (e - e^{\text{opt}})}{e^{\text{opt}}}.
\end{aligned}
$$

For finding the optimal solution, the NLPQLP algorithm of the pyOpt package has been used [18]. This algorithm is gradient-based and, in order to leverage the most out of the algorithm, NASTRAN has been used to compute structural responses and their derivatives. The algorithm is able to converge in less than twenty iterations as illustrated via Figure 13.

If optimizing just mass, a minimum of 5.2 kg was found. For the opposite case, i.e., solely minimizing effort, the level of manufacturing effort is 33%. For considering both mass and effort simultaneously, a mass of 5.9 kg and an effort level of 41% was obtained. Even more interesting, the optimization algorithm adopted the A-pillar, such that it suits for both goals of having a lightweight structure at minimal expenses. This is illustrated by Figure 14, where obviously the moment of inertia of the profile was enlarged. This did yield a structure displaying higher bending stiffness at lower mass.

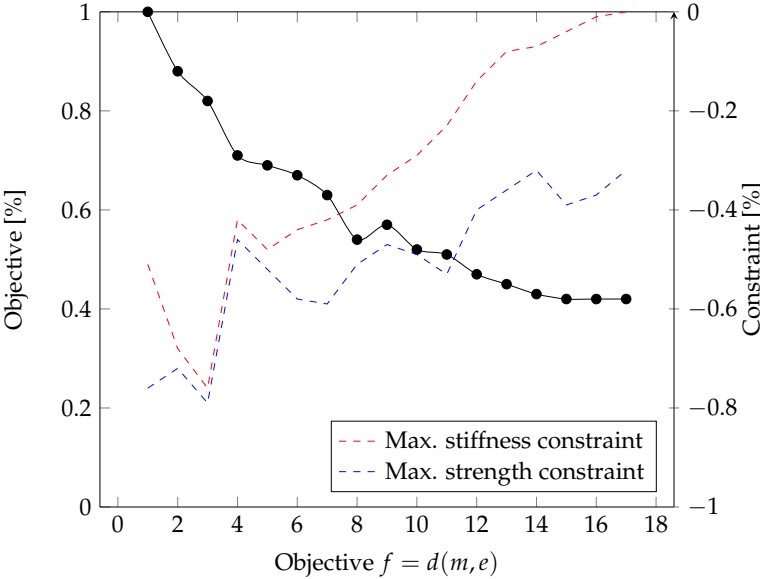

**Figure 13.** Convergence plot.

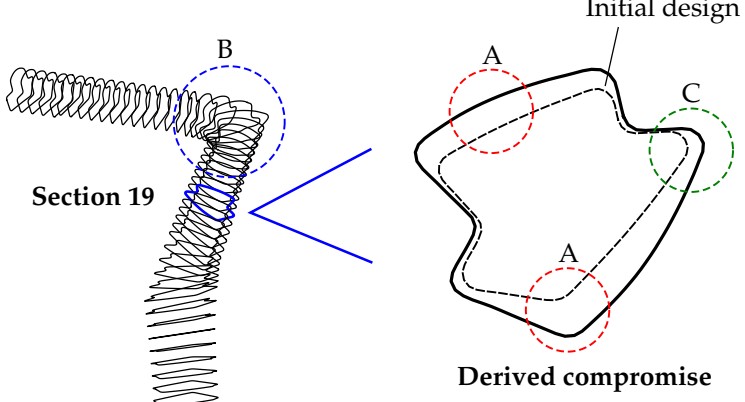

**Figure 14.** Illustration of the optimal compromise for the braided A-pillar.

For also honoring manufacturing, the circumferences of each profile were outbalanced with the chosen braiding angle so as to have reasonable manufacturing parameter values. This is depicted in Figure 15, where Figure 15a shows the course of braiding angle $\varphi(s)$ along the elongation axis $s$, Figure 15b the braider yarn width $w(s)$ and Figure 15c the resulting manufacturing effort for each profile.

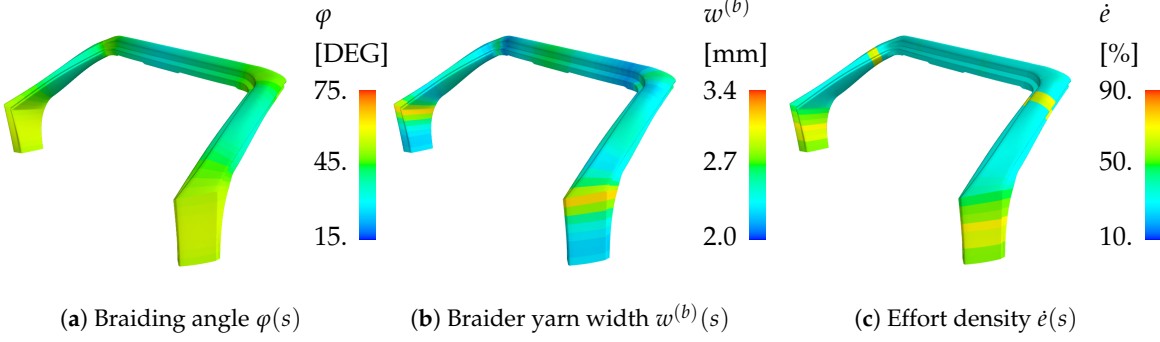

(**a**) Braiding angle $\varphi(s)$      (**b**) Braider yarn width $w^{(b)}(s)$      (**c**) Effort density $\dot{e}(s)$

**Figure 15.** Design variables $\varphi$ and $w$ as well as the response of effort model.

The distribution of effort per profile $\dot{e}(s)$ is the direct product of the MEM. Thus, the overall level of effort needs to be computed by

$$e = \frac{1}{l} \int_l \dot{e}(s)\mathrm{d}s. \tag{9}$$

By considering this, the designer is hence able to trace back which change did cause which level—or, more precisely, for high levels of manufacturing effort density $\dot{e}$, the designer is not only able to localize the region of interest, but moreover to query from the manufacturing effort model which set parameters did cause this level and how to counteract. This will be discussed in Section 4.2.

### 4.2. Computer Added Evaluation Capabilities

The developed MEM can moreover be directly used during designing. This enhances the design process since it allows direct feedback on level of producibility. In addition, the MEM is able to provide reasons for the conducted elaborations and advice for optimally improving the design. In order to enable this, the structure that will be braided is discretized by B-splines per profile, as depicted by Figure 16.

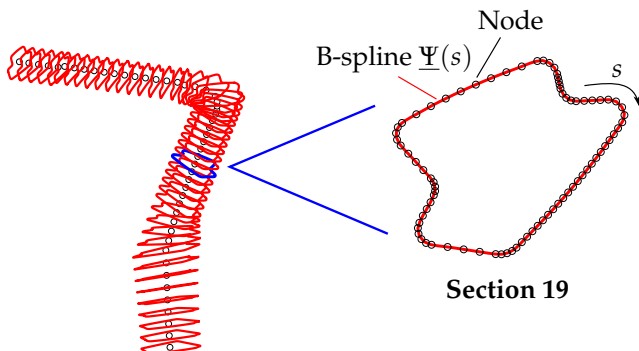

**Figure 16.** B-spline discretization of A-pillar.

Figure 17 shows the result of the analysis by the MEM per profile. As can be observed, the MEM passes, along with the level of effort, the reason for having determined this level and advice on how to improve the design. As can be observed, the curvature of the A-pillar (see region B highlighted by the blue circle in Figure 14) interacts with the braiding angle such that the minimum value is dependent on circumference, number of bobbins and curvature radius.

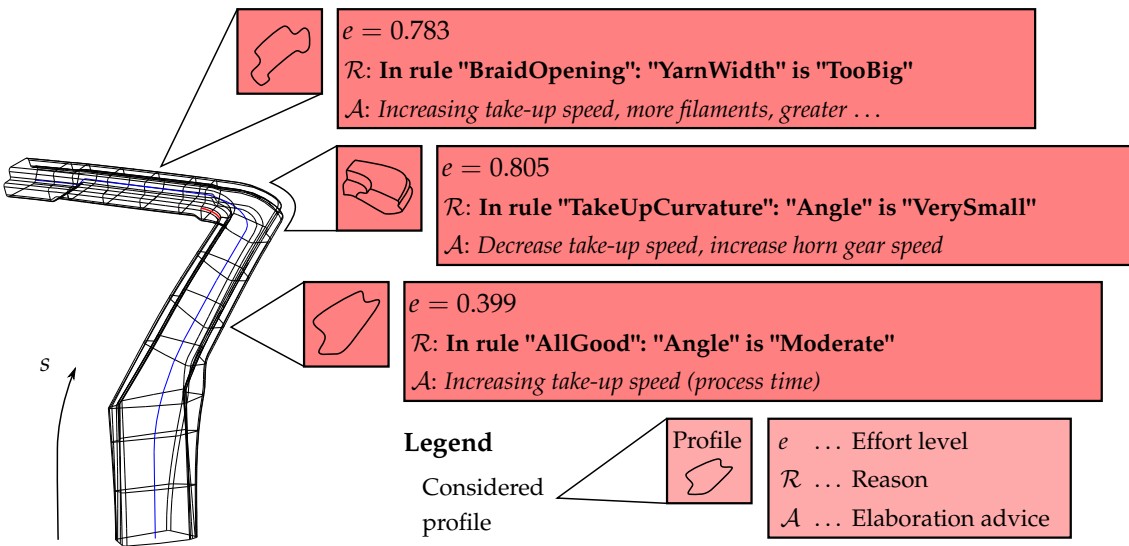

**Figure 17.** CAE capabilities of the proposed approach.

The MEM is able to consider this interaction and—as being part of the optimization frame—is moreover able to predict optimal configurations. As given with Figure 17, the MEM indicated that the braiding angle is right at the minimum for the initial case. The optimization was however able to reduce the level of 81% down to a moderate level, by adopting the braiding angle, the curvature in tandem with the profile shape. This can be comprehended by observing that the braiding yarn with is within its reasonable range (see Figure 15b, causing the effort to drop as illustrated by Figure 15c. Due to the additional responses and advice of the MEM, this could have been realized by a designer as well.

## 5. Discussion

Via narrow artificial intelligence, human expertise on manufacturing costs and restrictions of composite braiding can be emulated. This herein allowed to evaluate manufacturing effort for a given design, where effort was defined as a general measure. Moreover, this knowledge-based system was expanded by the capability of providing reason and advice. By doing so, the effort model remains numerically efficient and, therefore, fast in execution with negligible computation times. However, it was enriched, such that it can be utilized during the design of braided composite structures, where the responses of this intelligence allows goal-orientated and live design improvements, which makes it advantageous over numerically intensive methods like direct process simulations.

Additionally, a manufacturing effort model is straightforward enough to be included in an optimization frame. This yields a holistic approach, facilitating the derivation of true optimal compromises. These compromises are superior because of simultaneous consideration of structural mechanics and manufacturing aspects. Hence, this is opposed to first incrementally improving a design towards mass and then adopting it, such that it can be produced, where the latter may scrutinize most of the lightweight gains. Consequently, the proposed approach is able to consider aspects of multiple disciplines at the same time, so as to strike an optimal compromise.

When comparing the methods, as introduced in Section 1.2, it becomes obvious that the proposed approach is superior in regard to computational expenses, if in contrast with direct process simulations. However, it is of course limited to the knowledge reflected within its knowledge base. This is due to the fact that knowledge can not be extrapolated beyond the boundaries of its domain. However, it is by far more complete than any analytical approach because it captures complex interactions in-between process parameters, which might even not be explicitly expressible.

Last but not least, it shall be noted that the approach proposed herein is of a general nature and could be applied to any given manufacturing process and is implemented into CAE tools or optimization frames with ease.

## 6. Materials and Methods

Upon request, the author shares all details of the approach, as well as the code to realize the manufacturing effort model. The optimization strategy does not need to be published, since it is already published and described by [18].

## 7. Conclusions

Herewith, a numerical efficient approach of capturing manufacturing restrictions and costs based on narrow artificial intelligence has been proposed. It is further extended, such it provides reasons for having determined a level of effort. Moreover, it advises on how to improve the design in the most optimal fashion. The approach has been applied within an optimization frame, where mass and manufacturing effort of an automotive structure was successfully optimized simultaneously. Lastly, its CAE capabilities allowing the derived effort model to be used live during design.

**Funding:** The findings presented herein were produced without any external funding. However, initial research work was generated within the author's doctoral studies at the Technical University Munich—Institute of Lightweight Structures; more particularly by research work on MAI Design project as being part of the MAI Carbon cluster and funded by the Federal Ministry for Training and Research (BMBF).

**Acknowledgments:** The author would like to again thank the MAI Carbon cluster for giving me the possibility to contribute to research matters on design and analyses of carbon structures.

**Conflicts of Interest:** The author declares no conflict of interest. Moreover, no other party such as funders and the like had a role in the design of the study; in the collection, analyses, or interpretation of data; in the writing of the manuscript, or in the decision to publish the results.

## Abbreviations

The following abbreviations are used in this article:

| | |
|---|---|
| CAE | Computer Added Engineering |
| FEM | Finite Element Method |
| FIS | Fuzzy Inference System |
| KBS | Knowledge-Based System |
| MDDO | Multi-Disciplinary Design Optimization |
| MEM | Manufacturing Effort Model |
| NASTRAN | NASA Structural Analysis System (FFE software) |
| NLQLP | Gradient-based Nonlinear Programming Aalgorithm |

## Appendix A. Design Space and Parametrization

Parametrization of profiles is given with Figure A1 and the following Table A1 provide an overview on the complete associated design space.

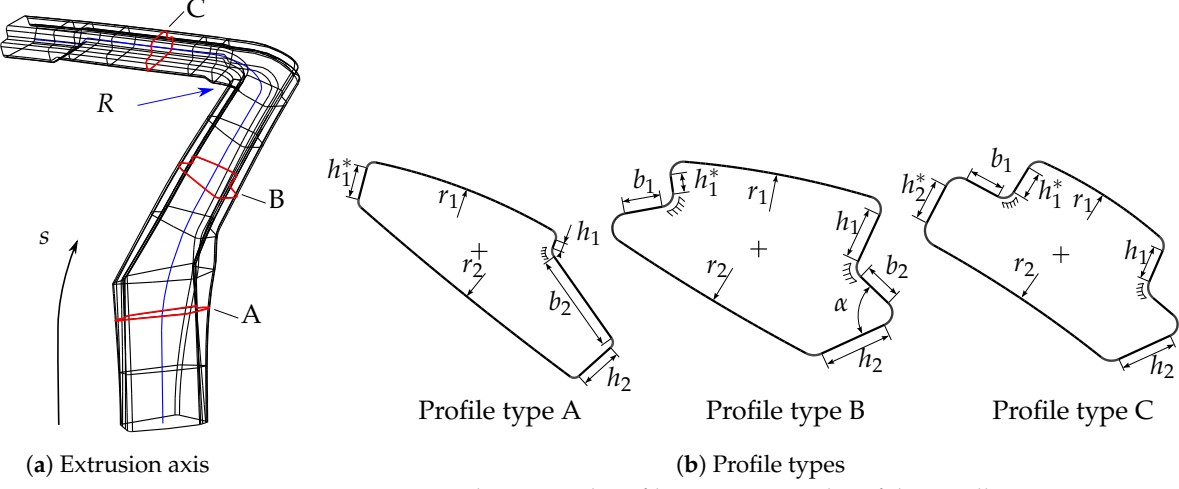

(**a**) Extrusion axis            (**b**) Profile types

**Figure A1.** Extrusion axis and associated profile types A, B and C of the A-pillar.

**Table A1.** Definition of the design space - the shape variables of the profiles.

| $\underline{x}$ | $\underline{x}^l$ | $\underline{x}^u$ | Unit | ID |
|---|---|---|---|---|
| $b_1^{(B)}$ | $-10.$ | $20.$ | mm | 1 |
| $b_2^{(B_1)}$ | $-5.$ | $15.$ | mm | 2 |
| $b_2^{(B_2)}$ | $-5.$ | $30.$ | mm | 3 |
| $b_2^{(B_3)}$ | $-5.$ | $10.$ | mm | 4 |
| $h_1^{(B_1)}$ | $-3.$ | $20.$ | mm | 5 |
| $h_1^{(B_2)}$ | $-2.$ | $20.$ | mm | 6 |
| $h_1^{(B_3)}$ | $-3.$ | $20.$ | mm | 7 |
| $h_2^{(B_1)}$ | $-10.$ | $15.$ | mm | 8 |
| $h_2^{(B_2)}$ | $-10.$ | $15.$ | mm | 9 |
| $h_2^{(B_3)}$ | $-10.$ | $20.$ | mm | 10 |
| $\alpha^{(B_1)}$ | $-10.$ | $20.$ | DEG | 11 |
| $\alpha^{(B_2)}$ | $-10.$ | $50.$ | DEG | 12 |
| $\alpha^{(B_3)}$ | $-10.$ | $20.$ | DEG | 13 |
| $r_1^{(B_2)}$ | $-2000.$ | $2000.$ | mm | 14 |
| $r_1^{(B_3)}$ | $-2000.$ | $2000.$ | mm | 15 |
| $r_2^{(B_2)}$ | $-4000.$ | $4000.$ | mm | 16 |
| $r_2^{(B_3)}$ | $-1000.$ | $1000.$ | mm | 17 |
| $b_1^{(C)}$ | $-10.$ | $20.$ | mm | 18 |
| $h_1^{(C)}$ | $-7.$ | $10.$ | mm | 19 |
| $h_2^{(C)}$ | $-7.$ | $10.$ | mm | 20 |
| $r_1^{(C)}$ | $-1000.$ | $1000.$ | mm | 21 |
| $r_2^{(C_1)}$ | $-4000.$ | $2000.$ | mm | 22 |
| $r_2^{(C_2)}$ | $-4000.$ | $3000.$ | mm | 23 |
| $R$ | $-1.$ | $0.5$ | - | 24 |
| $t_1$ | $2.$ | $8.$ | mm | 25 |
| $t_2$ | $2.$ | $8.$ | mm | 26 |
| $t_3$ | $2.$ | $8.$ | mm | 27 |
| $t_{P,1}$ | $0.001$ | $0.6$ | mm | 28 |
| $t_{P,2}$ | $0.001$ | $0.6$ | mm | 29 |
| $t_{P,3}$ | $0.001$ | $0.6$ | mm | 30 |
| $t_{P,4}$ | $0.001$ | $0.6$ | mm | 31 |
| $\varphi_1$ | $15.$ | $75.$ | DEG | 32 |
| $\varphi_2$ | $15.$ | $75.$ | DEG | 33 |
| $\varphi_3$ | $15.$ | $75.$ | DEG | 34 |
| $\varphi_4$ | $15.$ | $75.$ | DEG | 35 |
| $\varphi_5$ | $15.$ | $75.$ | DEG | 36 |

## Appendix B. Overview on Response Surfaces of the Braiding Manufacturing Effort Model

With the following Figure A2, the reader shall be able to visually check continuity of the most relevant response surfaces.

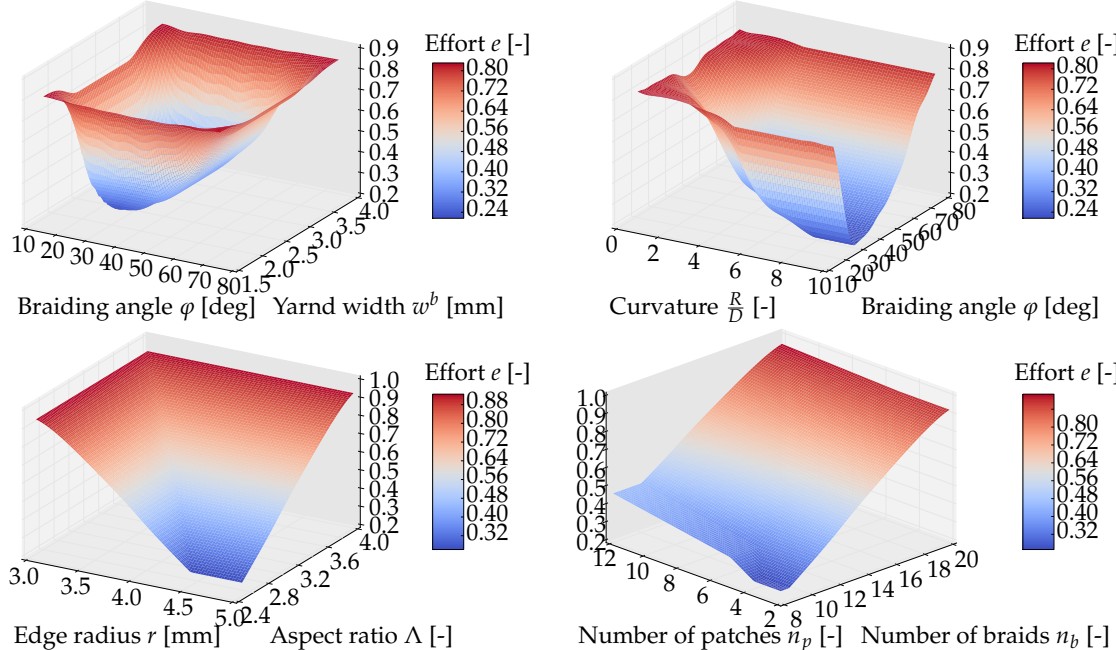

**Figure A2.** Response surfaces of the manufacturing effort model over different process variables.

## Appendix C. Graph of Knowledge-Basis

Via Figure A3, the core of the braiding manufacturing effort model is given. It illustrates the knowledge-base in the form of a network of logical elements linking input variables on the left with output effort, reason and advice.

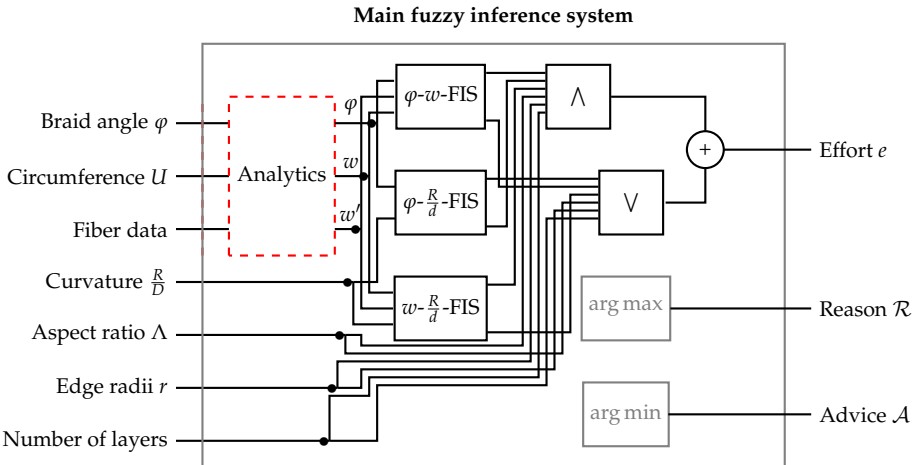

**Figure A3.** Illustration of the fuzzy inference system for the braiding manufacturing effort model.

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
