# Peer review of "Enabling Composite Optimization through Soft Computing of Manufacturing Restrictions and Costs via a Narrow Artificial Intelligence"

_jcs, doi:10.3390/jcs2040070_

Round 1
Reviewer 1 Report
First of all, I would like to congratulate good paper. This contribution showcases a very interesting idea to achieve composite optimization via a weak artificial intelligence emulating human expertise on composite manufacturing. Overall this paper is very well written and the topic it addressed is very relevant to the scope of Journal of Composites Science. I would recommend publication of this paper. See below some minor comments.
(1) Line 138: “can be directly using” should be “can be directly used.”
(2) Line 140: “for optimally improve” should be “for optimally improving”.
Author Response
First and foremost, thanks for your feedback and valued inputs. Both comments were directly resolved and will be correctly included the latest version.
Best Regards and thanks a lot for the fast review
Markus Schatz
Reviewer 2 Report
This manuscript presents the development and use of a numerical tool (weak artificial intelligence) in order to optimize composite structures manufacturing and costs. The topic is quite interesting and it may be useful for other researchers. The structure of the paper is well organized and the overall paper tells a logical story with a concrete conclusion. It is suitable for publication in Journal of Composites Science. Nevertheless, the Introduction section should be enhanced with additional works related to the optimization of composite structures using numerical approaches.
Author Response
Dear reviewer,
first and foremost, thanks for your feedback and valued inputs. The comments you made were sharp and therefore regarded of high importance. I instantly did include three additional citations on the matter of direct simulation of composite manufacturing. They namely are:
Pickett, A.K., Sirtautas, J. and Erber, A.: Braiding Simulation and Prediction of Mechanical Properties
Chen, C. S., Chen, T. J., Chen, S. C. and Chien, R. D.: Optimization of the injection
molding process for short-fiber-reinforced composites
Mackerle, J. : Finite element analyses and simulations of manufacturing processes of composites
and their mechanical properties: a bibliography
I think that your comment was more than important and i am happy to have included these three papers, since they underpin the difficulty of composite process simulation and on top the expanses in case these simulations shall serve as input for a optimization frame.
In case you have further doubts, please do not hesitate to write me at any given time.
Best Regards and thanks a lot for the fast review
Markus Schatz
Reviewer 3 Report
The authors submitted the article entitled “Enabling Composite Optimization Through Soft Computing Of Manufacturing Restrictions And Costs Via A Narrow Artificial Intelligence”. I recommend that the paper could be accepted after minor revisions. My main comments and questions are as follows:
1. The authors did quite a lot of simulations and illustrations. It’s necessary to connect or address the results to some real examples. Having these connections, we can utilize their results for structural designs.
2. The authors should also discuss about the properties of the composites, such as thermal and mechanical properties.
3. Sine there’re several optimization methods, the authors should address/compare the effectiveness of their method to others.
4. The authors should enrich the Introduction with more piratical examples.
5. The references are also insufficient to reflect the research background.
Author Response
Dear reviewer,
first and foremost, thanks for your feedback and valued inputs. The comments you made were sharp and therefore regarded to be of high importance.
As I herewith outline, the five comments were acknowledged as follows:
Section 4: All made simulations and optimizations were later considered by BMW and AUDI for their design of braided structures. Moreover, the work performed herein enabled the update of the A-pillar used by the company RODING. They did actually built this Roding Roadster car. For that reason, i am more than happy that you made this remark, since the first version did not made this obvious. I added more statements on that topic in section 4. In addition more optimization results were provided, as well as discussed and put into an industrial frame. Last but not least, the sub-sections were re-arranged, so as to make it clear how the manufacturing effort model can be used. So thanks a lot for your remark.
Section 1.4: This actually is a really good remark. In order to provide some values i added a plot showing the effective stiffness of braid in braiding axis over braiding angle and braiding width. Furthermore, mechanical and thermal properties are discussed now.
Section 5: Again, thanks for commenting this. It is more than true that this has not been addressed in the first version. The conclusion section was revised and more on this regard is included.
Section 1.3: With this new section, another relevant application example is discussed. This one is derived from aeronautics; the proper structure and was developed based on needs of Premium AEROTEC, Airbus and TU Munich. The research on this matter has been published and referenced here as well.
Section 1.2: Three additional citations were introduced on the matter of direct simulation of composite manufacturing. They were regarded important, since they underpin the difficulty of composite process simulation and on top the expanses in case these simulations shall serve as input for a optimization frame. They namely are:
Pickett, A.K., Sirtautas, J. and Erber, A.: Braiding Simulation and Prediction of Mechanical Properties
Chen, C. S., Chen, T. J., Chen, S. C. and Chien, R. D.: Optimization of the injection
molding process for short-fiber-reinforced composites
Mackerle, J. : Finite element analyses and simulations of manufacturing processes of composites
and their mechanical properties: a bibliography
I think that your comments were more than important and i am happy to have included your comments in the current version. In case you have further doubts, please do not hesitate to write me at any given time.
Best Regards and thanks a lot for the fast review
Markus Schatz